# Vacuolar Proton-Translocating ATPase May Take Part in the Drug Resistance Phenotype of Glioma Stem Cells

**DOI:** 10.3390/ijms25052743

**Published:** 2024-02-27

**Authors:** Martina Giambra, Andrea Di Cristofori, Francesca Raimondo, Roberta Rigolio, Donatella Conconi, Gaia Chiarello, Silvia Maria Tabano, Laura Antolini, Gabriella Nicolini, Miriam Bua, Davide Ferlito, Giorgio Carrabba, Carlo Giorgio Giussani, Marialuisa Lavitrano, Angela Bentivegna

**Affiliations:** 1PhD Program in Neuroscience, University of Milano-Bicocca, 20900 Monza, Italy; m.giambra1@campus.unimib.it (M.G.); a.dicristofori@campus.unimib.it (A.D.C.); 2GBM-BI-TRACE (GlioBlastoMa-BIcocca-TRAnslational-CEnter), University of Milano-Bicocca, 20900 Monza, Italy; francesca.raimondo@unimib.it (F.R.); gaia.chiarello@irccs-sangerardo.it (G.C.); giorgio.carrabba@unimib.it (G.C.); carlo.giussani@unimib.it (C.G.G.); marialuisa.lavitrano@unimib.it (M.L.); 3School of Medicine and Surgery, University of Milano-Bicocca, 20900 Monza, Italy; roberta.rigolio@unimib.it (R.R.); donatella.conconi@unimib.it (D.C.); laura.antolini@unimib.it (L.A.); m.bua@campus.unimib.it (M.B.); d.ferlito1@campus.unimib.it (D.F.); 4Neurosurgery, Fondazione IRCCS San Gerardo dei Tintori, 20900 Monza, Italy; 5Pathology, Fondazione IRCCS San Gerardo dei Tintori, 20900 Monza, Italy; 6Laboratory of Medical Genetics, Ospedale Maggiore Policlinico, IRCCS Ca’ Granda, 20122 Milan, Italy; silvia.tabano@unimi.it; 7Department of Pathophysiology and Transplantation, University of Milan, 20122 Milan, Italy

**Keywords:** glioblastoma, GBM, glioma stem cells, GSCs, chemoresistance, V-ATPase, bafilomycin A1, temozolomide, autophagy process

## Abstract

The vacuolar proton-translocating ATPase (V-ATPase) is a transmembrane multi-protein complex fundamental in maintaining a normal intracellular pH. In the tumoral contest, its role is crucial since the metabolism underlying carcinogenesis is mainly based on anaerobic glycolytic reactions. Moreover, neoplastic cells use the V-ATPase to extrude chemotherapy drugs into the extra-cellular compartment as a drug resistance mechanism. In glioblastoma (GBM), the most malignant and incurable primary brain tumor, the expression of this pump is upregulated, making it a new possible therapeutic target. In this work, the bafilomycin A1-induced inhibition of V-ATPase in patient-derived glioma stem cell (GSC) lines was evaluated together with temozolomide, the first-line therapy against GBM. In contrast with previous published data, the proposed treatment did not overcome resistance to the standard therapy. In addition, our data showed that nanomolar dosages of bafilomycin A1 led to the blockage of the autophagy process and cellular necrosis, making the drug unusable in models which are more complex. Nevertheless, the increased expression of V-ATPase following bafilomycin A1 suggests a critical role of the proton pump in GBM stem components, encouraging the search for novel strategies to limit its activity in order to circumvent resistance to conventional therapy.

## 1. Introduction

High-grade gliomas are malignant primary brain tumors, among which glioblastoma (GBM) is the most frequent. They account for 69% of all gliomas and for 12–15% of all primary brain tumors [1,2]. The current treatment consists of maximal safe surgical resection followed by concomitant chemo-radiotherapy [3,4,5]. First-line chemotherapy involves the administration of temozolomide (TMZ), an alkylating agent which induces DNA damage in tumoral cells. This action is particularly effective in patients with an increased methylation level in the promoter of the DNA-repairing enzyme O6-methylguanine-DNA methyltransferase (*MGMT*), thus configuring *MGMT* as a prognostic and predictive marker in GBM [6,7]. Patients with methylated *MGMT* are known to have the best overall survival among patients with GBM [6,7,8]. On the contrary, patients with a low level of methylation of *MGMT* are known to have the worst overall survival, and this is due to a lower efficiency of the alkylating agents [5,7]. Although the recent addition of tumor-treating fields (TTF) as a first-line therapy has been proven to increase tumor local control, tumor recurrence is almost always expected, causing a poor outcome for patients [9]. Tumor recurrence is a consequence of a tumor’s high resistance to chemo-radiotherapy [10]. Several therapeutic options are under investigation, and future treatment for patients with GBM will probably involve tailored therapies based on molecular tumor features [11,12].

The vacuolar proton-translocating ATPase (V-ATPase) is a membrane-bound, multi-subunit enzyme that is involved in maintaining the intracellular pH through the ATP-dependent transport of H+ from the cytoplasm to the extracellular environment and is also involved in the acidification of endosomes and lysosomes in eukaryotic cells [13,14,15,16]. It is made of a membrane-embedded V0 sector, which regulates proton permeability, and an enzymatic V1 ATPase sector [14,16,17]. In carcinogenesis, anaerobic glycolysis takes over as the primary metabolic pathway for neoplastic cells. This activity produces lactates and H+, although it can readily provide ATP through the consumption of glucose [17]. Lactate overproduction would cause the intracellular pH to drop and, as a result, trigger apoptotic events that would kill cancerous cells [18]. Under these conditions, it has been demonstrated that V-ATPase plays a critical function in restoring intracellular pH levels that support cellular self-renewal and in counteracting the excess H+ generated [17]. Moreover, several authors have proposed that V-ATPase may be involved in drug resistance by participating in drug extrusion from the intracellular environment [13,15]. Lastly, it is also thought that a high H+ efflux causes the extracellular matrix to become acidic. This condition allows the protonation of antitumoral drugs and, therefore, a lower entry into tumor cells and, consequently, a lower effectiveness of chemotherapy [15,19,20]. The inhibition of V-ATPase is supposed to be a therapeutic way of blocking the tumoral metabolic machinery that leads cancerous cells to apoptosis and overcoming drug resistance [21].

GBM originates from stem cells that are called glioma stem cells (GSCs) and that are thought to be responsible for tumor self-maintenance, spreading, and resistance to adjuvant treatments [22,23]. Many biological mechanisms are thought to be involved in GBM resistance to drugs and radiotherapy [23].

Thanks to previous studies, V-ATPase is known to be a key player in gliomagenesis, especially in isocitrate dehydrogenase 1 or 2 (*IDH1* or *IDH2*) wild-type gliomas [24,25,26,27,28]. Moreover, V-ATPase is supposed to participate in the drug resistance that characterizes high-grade gliomas, and it may be involved in the acidification of the glioma environment [26,29]. Finally, the expression of V-ATPase increases proportionally with the tumor’s grade [29]. Although some experiences are reported in the literature, it is still poorly known how V-ATPase inhibitors could be administered to patients and what role they could play in a neuro-oncological setting (administrated as single chemotherapy agents or as combined or adjuvant therapy). For all the above, finding an in vitro model for V-ATPase inhibition would be beneficial to patients with GBM in order to potentially plan a translation from the bench to the bed of the patient. This would be especially important for those patients with a low level of *MGMT* methylation who, nowadays, lack an alternative therapeutic option to TMZ as a first-line treatment.

In this view, the aim of our work is to analyze the effects of V-ATPase inhibition by bafilomycin A1 (Baf) on patient-derived GSC lines in order to characterize the antitumoral effects, toxicities, and potential applications in vivo. Our data question the potential use of Baf alongside TMZ because of both the absence of efficacy in reverting TMZ resistance and the already significant presence of a strong cytotoxicity of the drug in the GSC lines tested.

## 2. Results

### 2.1. The Sensitivity of Glioma Stem Cell Lines to Temozolomide In Vitro Is Largely Independent of MGMT Promoter Methylation

TMZ represents the first-line chemotherapy drug in GBM treatment. In order to assess the well-known resistance of the stem component to this standard drug [30], cytotoxicity assays were performed on three patient-derived glioma stem cell lines (GSC7, GSC22, and GSC23). The cells were treated with 100 and 200 µM TMZ for 48 and 72 h. No concentrations of TMZ were able to significantly reduce cell viability at both time points (Figure 1).

The methylation status of the *MGMT* promoter is a favorable prognostic marker in GBM since it is correlated to chemosensitivity to the alkylating agent TMZ. In order to evaluate if the TMZ resistance observed in the GSC lines was correlated to the *MGMT* methylation status, a quantitative CpG methylation analysis was performed by pyrosequencing. The analysis was executed both on the tumor core (TC) bulks and on the derived GSC lines (Table 1).

The methylation status of the *MGMT* promoter was preserved between the two types of samples, even if differences in the methylation percentage were seen in the TC-GSC pair number 7 and 22. Nevertheless, the methylation status did not correlate with the response to TMZ. Indeed, despite all the GSC lines being resistant to TMZ, two of them (GSC22 and GSC23) had an *MGMT* promoter that was hyper-methylated, which would suggest a sensitive phenotype which, instead, was not observed.

These results showed that the methylation status of the *MGMT* promoter is not the exclusive mechanism that regulates the chemoresistance of GBM to alkylating agents like TMZ. 

### 2.2. V-ATPase Is Overexpressed in Tumor Core Bulks and Glioma Stem Cell Lines

Since GSC lines are resistant to TMZ, finding new therapeutic targets hampering GBM progression and supporting TMZ therapy is necessary. Previous studies have shown that the vacuolar proton-translocating ATPase (V-ATPase) could be a good therapeutic target for treating GBM given its role in gliomagenesis and its overexpression in high-grade gliomas [25,28,29]. Therefore, the expression of ATP6V1G1, a subunit of the V1 sector of the protons H+ transporting pump, was evaluated in our cell lines. ATP6V1G1 expression was assessed in the TC bulks by immunohistochemistry (IHC) using an anti-V-ATPase-G1 antibody. Even if all the samples expressed the protein, the levels of immunoreactivity were variable among them: TC7 had 70% of positive cells, TC22 40%, and TC23 80% (Figure 2a). The consultation of the Gene Expression Profiling Interactive Analysis 2 (GEPIA2) platform proved these results in a larger cohort of GBM tissues where tumor specimens were compared to normal brain tissues (Figure 2b,c), suggesting once again V-ATPase’s potential role as a therapeutic target.

We evaluated the V1G1 subunit’s expression in the derived GSC lines by a Western blotting analysis. No statistical difference was observed among the three GSC lines and the HCT116 colon cancer cell line, probably because of the wide variability observed in all the samples. The HCT116 colon cancer cell line was used as the positive control, since the subunit is known to be overexpressed also in colon cancers (Figure 3) [31]. GSC7 showed good accordance with the high level of expression of V1G1 in the TC sample. This did not occur for the other two samples.

Finally, we checked the correct localization of the proton pump by immunofluorescence experiments on GSCs. Antibody positivity was visualized both on the plasma membrane and on internal membranes (Appendix A). These results produced data in favor of the overexpression of the V-ATPase pump in the GSC lines.

### 2.3. Nanomolar Doses of Bafilomycin A1 Do Not Overcome Temozolomide Resistance in Glioma Stem Cell Lines

We wanted to evaluate if the inhibition of the V-ATPase pump could revert the TMZ resistance in the GSC lines. In this regard, we selected bafilomycin A1 (Baf) as an inhibitor of V-ATPase since it is the first-line molecule identified with this purpose and is efficient in a wide range of organisms at nanomolar concentrations [32]. Firstly, dose–response curves of nanomolar doses of Baf were tested for 48 h in order to identify a GSC line-specific sub-lethal dosage potentially effective in reverting TMZ resistance (Figure 4).

With this intent, we arbitrarily selected, for each GSC line, the highest dose of Baf where the percentage of cell viability was ≥90% (GSC7 10 nM, GSC22 5 nM, and GSC23 5 nM). The selected doses, as reported by a previous study in animal models, were considered to be safe [33]. Then, the GSCs were co-treated for 48 h with the selected dose of Baf and TMZ 100 µM. We used two different treatment plans of drug combinations to identify the better way to potentially recover the resistance to TMZ in the cell lines: simultaneous administration for 48 h (plan 1); and 24 h Baf and then TMZ addition up to 48 h of treatment (plan 2) (Figure 5). 

The results showed that the combination of sub-lethal doses of Baf with TMZ did not overcome the resistance of the GSC lines to the alkylating agent in either of the two administration plans used. Subsequently, in order to assess whether the inefficiency of the proposed treatments was due to a low Baf dose, we repeated the treatments, boosting the dose of Baf to 20 nM (Figure 6).

Generally, the viability of all the GSC lines was reduced after Baf administration, both by itself or combined with TMZ in the two plans of treatment, while resistance to TMZ was confirmed for all of them. In particular, GSC7 showed a significant sensitivity only to the combination of Baf and TMZ with plan 2. Conversely, GSC22 and GSC23 showed a significant sensitivity to Baf administered both alone and in the two combination plans. Interestingly, the combined plans 1 and 2 showed benefits in terms of significant reduction in the cell viability. Indeed, plan 1 was effective in GSC22 and GSC23, while plan 2 was effective in all three samples, compared to CTR DMSO. However, almost the same reduction in cell viability as that achieved by the combined treatments was obtained by administrating Baf alone.

### 2.4. Effect of Bafilomycin A1 Treatment on V-ATPase Protein Expression 

In order to investigate a possible role of Baf on V-ATPase protein expression, we assessed through Western blotting its level after 48 h of treatment. Interestingly, the protein level of the V1G1 subunit significantly increased at both of the doses used (20 nM and 40 nM) compared to the untreated condition (CTR DMSO) in the three GSC lines. In addition, no differences were highlighted between the two drug doses (Figure 7). 

### 2.5. Autophagy Pathway in Glioma Stem Cell Lines Is Blocked by Bafilomycin A1

Since it is known that, at micromolar doses, Baf inhibits the late stages of the autophagy pathway [33], we investigated this mechanism to explain the cytotoxic effect observed with nanomolar doses of Baf. We used LC3I and LC3II as the markers of autophagy, since, during autophagy, LC3I is converted to the lipidated form LC3II, which is subsequently degraded during the final step of the pathway [34]. The protein levels of LC3I and LC3II in the GSC lines after 20 nM and 40 nM Baf treatments for 48 h were compared to the untreated condition (CTR DMSO) (Figure 8a). 

Firstly, once again, a wide intra-sample variability was observed in all three GSC lines. At both drug doses, all the GSC lines showed an accumulation of LC3II in comparison to the loading control (vinculin). The results were statistically significant compared to CTR DMSO, indicating that treatment with Baf prevents LC3II degradation (Figure 8b). Secondly, LC3II/LC3I ratio increases were observed in all the GSC lines after exposure to the drug at both doses (Figure 8c). These results suggest that nanomolar doses of Baf inhibit the late stages of autophagy in GSC lines.

### 2.6. Bafilomycin A1 Induces Necrosis in Glioma Stem Cell Lines

A flow cytometry analysis was performed to assess cell death induced by 20 nM Baf as a single or combined treatment. For this analysis, we selected the combined treatment performed at different times (plan 2, as described before), since it is efficient in all three lines, as previously shown in our cytotoxicity assays. The percentage of viable, early-apoptotic, and late-apoptotic/necrotic cells was determined by Annexin V/Propidium Iodide staining. The basal level of viability (CTR DMSO) varied among the GSC lines, ranging from 69.8% to 77.8% (Figure 9). Furthermore, all the GSC lines confirmed their resistance to TMZ, showing a percentage of alive cells similar to that of their respective control (Figure 10).

In the TMZ-treated GSC7 and GSC22 samples, the cytotoxicity seemed to be related to late apoptosis/secondary necrosis; meanwhile, in the TMZ-treated GSC23 line, there was also a necrotic cell death component. The Baf treatment caused, in the GSC7 cells, mainly necrosis. This latter process was equally balanced with the late apoptosis/secondary necrosis observed in the GSC22 and GSC23 samples (Figure 9). Considering plan 2, the cytotoxic effects observed for each of the GSC lines seemed to be the result of the combination of the cellular death pathways caused by the two single drugs.

## 3. Discussion

GBM remains a tumor with a poor prognosis. Nowadays, progression-free survival and overall survival are achieved with maximal safe surgical resection followed by adjuvant concomitant chemo-radiotherapy and, finally, adjuvant chemotherapy and tumor-treating fields [9].

In this scenario, the median progression-free survival timeframe is 6.7 months and that of median overall survival 20.9 months [9]. Such poor chances of survival are due to the resistance of GBM to chemo-radiotherapy. In addition, the presence of a stem component inside GBM, naturally resistant to conventional therapies and responsible for tumor maintenance and progression, further complicates finding efficient therapies [35].

Nevertheless, an improvement in the median overall survival timeframe to 31.6 months is achieved by patients with hyper-methylation of the *MGMT* promoter. Indeed, the reduced expression of MGMT, the main chemoresistance factor of GBM, increases the responsiveness of these patients to TMZ [9,36]. The troubling phenomenon of TMZ resistance remains a long-standing issue since it has been demonstrated that MGMT is not the sole mechanism of drug resistance and that GSCs are the insidious component and the main character in this problem [37]. Therefore, survival data on patients sensitized to alkylating agents encourage researchers to find novel therapeutic approaches to overcome resistance to alkylating agents, eliminating both the entire tumor bulk and also GSCs [38,39].

In order to address this clinical challenge, using in vitro models based on glioma stem cells derived from patients could be a good opportunity to study what happens in clinics and represents a good way of conducting a first-line screening of new potential therapeutic approaches. Not to mention that cultures derived from patients maintain the heterogeneous scenario characteristic of the tumor in question, which is fundamental to understanding the cancer’s biology and finding effective strategies to eradicate it.

With these in mind, in this study we used three patient-derived glioma stem cell lines to test a new approach to overcome their resistance to the conventional first-line therapy for this cancer. We first showed that patient-derived GSCs were resistant to the standard treatment with TMZ, as had already been highlighted in the literature [35].

In all the GSC lines, the methylation percentage of the *MGMT* promoter increased in comparison to the tumor bulks in which the cells originated. Nevertheless, none of them were found to be sensitive to the drug in vitro. These data proved, once again, that *MGMT* is not the only mechanism of drug resistance in GBM, though it is the most studied. In fact, the cytotoxic action of TMZ is limited by an alteration in the mismatch–repair (MMR) complex, which, if not impaired, leads to cell cycle arrest and apoptosis, recognizing mis-paired DNA [40,41]. Another drug resistance mechanism might be related to the vacuolar proton-translocating ATPase (V-ATPase), a multi-subunit proton pump localized on both the cell membrane and intracellular organelles’ membranes. In previous works, V-ATPase has been demonstrated to be overexpressed in gliomas according to their grade and has been theorized to be involved in gliomagenesis and glioma aggressiveness [25,28,29]. Its physiological role is to control protons’ efflux through cellular membranes in order to maintain a normal intracellular pH and acidify the endosomes for their activation [42]. Through its activity, V-ATPase modifies the tumor microenvironment, consequently enhancing the drug resistance mechanism. On the one hand, with the intent of contrasting the excessive production of lactates caused by the glycolytic metabolism of tumor cells, the extracellular space becomes more acidic, which leads to the protonation and inactivation of anticancer drugs [17,20]. Also, V-ATPase can act as an ATP-binding cassette (ABC) transporter that hampers the delivery of drugs throughout the membrane [29,43]. Such premises are quite relevant since, in recent years, new research strategies have aimed to revert resistance to TMZ using so-called “TMZ enhancers”, drugs and molecules which contrast the resistance mechanism in order to increase the action of this first-line treatment [44,45]. Based on this, once we assessed the resistance of the patient-derived glioma stem cell lines to TMZ, our intent was to use Baf as a “TMZ-enhancer”, inhibiting V-ATPase. Our idea was supported by previous studies, where the combination of TMZ and a low dose of Baf (10 nM) resulted in reduced cellular viability in the GBM cells [29,46]. We first demonstrated the overexpression of V-ATPase in the GBM tissue of patients. Then, we verified whether the stem component preserved this characteristic and observed that only one sample had an increased level of V-ATPase both in the tumor tissue and in the derived stem line. In order to circumvent the well-known toxicity of Baf [47], the effects of customized dosages of Baf were then investigated in conjunction with TMZ, taking into account that each GSC line expressed varying quantities of the target. Contrary to what was expected from the literature, none of the lines showed a reduction in cell viability after the treatment, neither when administering the drugs simultaneously nor at different times. Even by increasing the dosages of Baf to 20 nM, the combination with TMZ was not effective. This result might be related to the fact that GSCs can develop multiple drug resistance strategies in order to reduce chemotherapy effects [48], while the effectiveness of the combined treatment reported in the literature, for example by Kanzawa, is obtained by treating the not-stem component of the tumor [46]. Also, we hypothesized that the inhibition of V-ATPase by nanomolar dosages of Baf might be deleterious for stem cells, which would increase the expression of the pump as a consequence. Overexpression of the target is one of the mechanisms adopted to escape the action of a drug [49]. Interestingly, we evaluated the expression levels of V-ATPase and observed a general increase in the pump’s level in all the lines after the treatment, supporting our hypothesis.

Even if this result might confirm the role of the V-ATPase pump in the drug resistance phenotype of GSCs, it also shows an important limit of using Baf. In fact, our in-depth investigations on the effect of Baf treatment on the glioma stem cells proved the cytotoxicity of the drug, even when used in low nanomolar dosages. According to a 2015 study by Yuan and colleagues, administering modest dosages of Baf selectively inhibits V-ATPase while having no effect on the autophagy pathway [33]. Here, we observed a global blockage of autophagy in all the samples after the treatment with 20 nM Baf. In addition, the activation of necrotic processes at different levels was evidenced. These data suggest two hypotheses: toxicity may be determined by the dosages of the treatment, so changing one or both factors could avoid an unspecified death; on the other hand, in our model, Baf may have had off-targets whose inhibition finally led to unexpected necrosis. In fact, it has been reported that Baf mainly induces apoptotic processes by increasing the production of ROS, not necrotic ones, as observed in our study [43].

In conclusion, our study questioned previous studies in which the combination of Baf with TMZ had been effective, maybe because they mostly focused on the not-stem component of GBM. On the one hand, our data pointed out the possible presence of other mechanisms of drug resistance conserved in the stem component of the tumor. On the other hand, the Baf treatment performed in this study cannot be applicable to a more complex model since the cytotoxicity of the drug was already evident in our cellular model. Nevertheless, the increased expression of V-ATPase after treatment with Baf may still suggest this protein’s role in the drug resistance phenotype. Therefore, finding alternative ways to inhibit it, either pharmacologically or genetically, will be part of our future work.

## 4. Materials and Methods

### 4.1. Patient-Derived Specimens

The collection of patient-derived specimens was part of a study approved by the ethic committee “Comitato Etico Monza e Brianza” (study number: 0031436—GLIODRUG-V, approved on 3 January 2020).

Tumor biopsies were collected from patients undergoing a craniotomy for a high-grade glioma at the Neurosurgery Unit of the Fondazione IRCCS San Gerardo dei Tintori (Monza Brianza, Italy) after informed consent had been obtained. The diagnosis of GBM was based on the recommendations of the 2021 WHO classification of CNS tumors [1]. The immuno-molecular phenotype and the genomic profile of each biopsy was published in Giambra et al.’s work in 2021 [50].

GSC7, GSC22, and GSC23 were isolated from the tumor biopsies thanks to a protocol developed in Giambra et al. (2021) where their stemness characteristic and genomic profile were assessed as well. The culturing conditions were also described in Giambra et al.’s work in 2021 [50].

### 4.2. Cytotoxicity Assay

Cell viability assays were performed after TMZ (Sigma-Aldrich, St. Louis, MO, USA) and Baf (Sigma-Aldrich) administration using Cell Counting Kit-8 (CCK-8) (Sigma-Aldrich), following the manufacturer’s instructions. TMZ and Baf were dissolved in DMSO (Euroclone S.p.A., Milan, Italy) at a 50 mM and 50 µM stock solution, respectively.

Briefly, the GSC lines were plated in a 96-well plate at a density ranging from 15,000 to 20,000 cells per well. Once 70% of growth confluency was achieved, the cells were treated with variable concentration of TMZ and Baf for 48 h or 72 h, alone or in combination (simultaneous administration for 48 h—plan 1; 24 h Baf and then TMZ addition up to 48 h of treatment—plan 2). The cells treated with DMSO (drug vehicle) were used as the control. At the end of the treatment, a CCK-8 solution was added to each well of the plate, and this was incubated at 37 °C for 4 h. The absorbance of each well was measured at 450 nm using a microplate reader (TECAN Infinite 200 Pro, Zürich, Switzerland). Cell viability percentage was calculated by dividing the treatment average absorbance by the control average absorbance and multiplying per 100. For each sample, at least three independent experiments were performed.

### 4.3. MGMT Methylation Analysis

The analysis of the tumor bulks was performed as reported in Marchi et al.’s work in 2019 [51], as part of the clinical diagnosis. Pyrosequencing experiments on the derived GSC lines were performed following the indications reported in Malacrida et al. in 2022 [52], using a Pyro Mark ID instrument in the PSQ HS 96 System (Biotage AB, Uppsala, Sweden), according to the manufacturer’s instructions. Raw data were analyzed using the Q-CpG software v1.0.9 (Biotage AB). For each sample, the methylation value represented the mean between two independent PCR and pyrosequencing experiments.

### 4.4. Immunohistochemistry

V1G1 subunit’s expression was evaluated by immunohistochemistry (IHC) on paraffin-embedded tissues of patients with GBM. The EnVision FLEX-High pH kit (Agilent Technologies, Santa Clara, CA, USA) with Dako Omnis (Agilent Technologies) was used following the manufacturer’s instructions. Briefly, the automated protocol involves a first cycle of deparaffinization followed by antigenic heat unmasking with a high pH EnV Flex TRS reagent and distilled water (dH2O). After the incubation of the primary antibody AntiV-ATPase-G1 (D5) (Santa Cruz Biotechnology, Santa Cruz, CA, USA, dilution 1:100) and the blockage of endogenous enzymes with Env Flex Perox/Blocking Agent, the labeled polymer was supplied (EnVision FLEX/HRP). Then, the chromogenic substrate EnVision Flex Substrate was added, followed by nuclear counterstaining with hematoxylin. Images were acquired with an Olympus microscope and the CellSens [Ver.3.1] IMAGING SOFTWARE (Evident Scientific Olympus, Olympus Corporation, Tokyo, Japan). The percentage of positive cells was calculated by counting the number of positive cells out of 100 cells in three independent sections.

### 4.5. Gene Expression Profiling Interactive Analysis 2 Platform Consultation

The Gene Expression Profiling Interactive Analysis 2 (GEPIA2) (http://gepia.cancer-pku.cn, Zhang Lab, Peking University, Beijing, China, last access date: 30 May 2023) [53] platform was used to evaluate the expression of the V1G1 subunit in GBM. The platform exploits RNA sequencing data from The Cancer Genome Atlas (TCGA) and Genotype Tissue Expression (GTEx) projects in order to compare V1G1 expression in GBM and in normal brain tissue. RNA sequencing data were analyzed using the parameter log10 (TPM+1) (transcripts per kilobase of exon model per million mapped reads).

### 4.6. Western Blotting

V1G1 and LC3I/II protein levels were investigated both in the untreated condition and after 48 h of treatment with 20 nM and 40 nM of Baf in the GSC7, GSC22, and GSC23 lines. The cells treated with DMSO (drug vehicle) were used as the control. Alpha-tubulin and vinculin were used as the loading control. Briefly, after the extraction, the same amount of proteins for each condition was separated by electrophoresis using 4–12% acrylamide Bolt Bis-Tris gels (ThermoFisher Scientific, Waltham, MA, USA), for the V1G1 experiments, and 14% gradient Trys-Glycin precast gel (Invitrogen, ThermoFisher Scientific), for the LC3I/II experiments, and transferred on nitrocellulose membrane for the V1G1 experiments and on a PVDF membrane for the LC3I/II experiments. The membranes were incubated with the following primary monoclonal antibodies: antiV-ATPase-G1 (D5) produced in mice (Santa Cruz Biotechnology, dilution 1:100), LC3I/II produced in rabbits (Cell Signaling Technology, Danvers, MA, USA, dilution 1:1000), anti-α-tubulin produced in rabbits (Cell Signaling Technology, dilution 1:1000), and anti-vinculin V284 produced in mice (Merk Millipore, Burlington, MA, USA, dilution 1:5000). The secondary antibodies were the following: anti-mouse IgG-HRP produced in goats (ThermoFisher Scientific, dilution 1:10,000) and anti-rabbit IgG-HRP produced in monkeys (ThermoFisher Scientific, dilution 1:10,000) were used in the V1G1 experiments; and anti-rabbit NA934V (Sigma-Aldrich, dilution 1:5000) and anti-mouse NA931 (Sigma-Aldrich, dilution 1:5000) were used in the LC3I/II experiments. Detection was carried out through an enhanced chemiluminescence assay (ECL). Images were acquired with an Amersham Imager 600 (GE Healthcare, Milan, Italy) and Gbox iChem, using the software GeneSys [Ver.1.4.0.0]. The bands were finally quantified with ImageQuantTL (GE Healthcare). For each condition, three different biological replicates were analyzed.

### 4.7. Immunofluorescence

Immunofluorescence was performed on GSCs, and the cellular localization of the V-ATPase proton pump was investigated using a specific antibody for its V1G1 subunit. AntiV-ATPase-G1 (D5) (Santa Cruz Biotechnology, dilution 1:100) was used as the primary antibody; Alexa Fluor 488-conjugated goat anti-mouse IgG green (Invitrogen, ThermoFisher Scientific, dilution 1:200) was used as the secondary antibody. The slides were mounted with the mounting medium with DAPI (Fluka Analytical, Sigma-Aldrich) and observed with a Radiance 2100 confocal fluorescence microscope (Bio-Rad, Hercules, CA, USA). The images were acquired using the Kalman filter, which allows for a good reduction of background noise. The overlay or merge of the images obtained was performed with Adobe Photoshop CS6 Extended version 13.0.

### 4.8. Flow Cytometry

In order to assess cell death after 48 h of treatment with 100 μM of TMZ alone, 20 nM of Baf alone, and in combination at a deferred time, the FITC-conjugated Annexin V-Propidium Iodide (PI) apoptosis detection kit (BD Biosciences, Buccinasco, Italy) was used according to the manufacturer’s instructions. Cells treated with the highest concentration of DMSO (drug vehicle) for each experimental setting were used as the controls. Briefly, cells were seeded in six-wells plate at a density ranging from 400,000 to 600,000 cells per well; once 70% confluence was achieved, the treatment was administered. At the end of the treatment, both the floating and attached cells were harvested, washed with PBS, and, finally, suspended in a binding buffer containing Annexin V and PI. After 15 min of incubation in the dark, the presence of apoptotic cells was analyzed by the FACSCantoI flow cytometer (BD Biosciences). At least 10,000 events/sample were acquired in an operator-defined region on a forward scatter–side scatter dot plot. The relative percentage of live/dead cells was assessed on an Annexin-V/PI dot using the FacsDiva software [Ver.5.0.3].

### 4.9. Statistical Analysis

In the cytotoxicity assays, the data were represented as the mean ± S.D. of at least three independent experiments; a paired *t*-test was performed as the statistical analysis on the raw data. Gene expression data from the TCGA and GTEx databases using the GEPIA2 platform were analyzed using a one-way ANOVA. In Western blotting, the data were represented as the mean ± S.D. of three different biological replicates; a paired *t*-test was performed as the statistical analysis of ratios. The data resulting from the analysis of flow cytometry were represented as the mean ± S.D. of three independent experiments; a paired *t*-test was performed as the statistical analysis on the raw data. In all the experiments, *p* < 0.05 was considered the minimum statistically significant *p* value; in the GEPIA2 analysis, *p* < 0.01 was considered the minimum statistically significant *p* value.

## Figures and Tables

**Figure 1 ijms-25-02743-f001:**
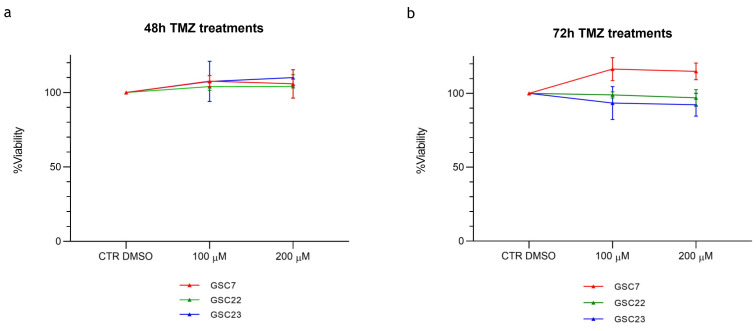
Cytotoxicity assay by Cell Counting Kit-8 (CCK-8) of glioma stem cell lines (GSC) 7, 22, and 23 treated for 48 h (**a**) and 72 h (**b**) with temozolomide (TMZ) (100 and 200 µM). Data are represented as the mean percentage of three independent experiments ± S.D. compared to a control (CTR DMSO, vehicle control) arbitrarily set to 100%. The statistical analysis was performed on raw data.

**Figure 2 ijms-25-02743-f002:**
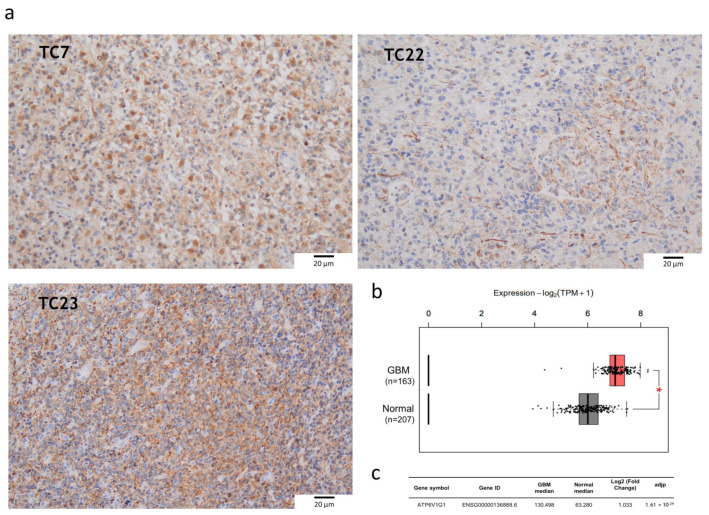
Expression of V-ATPase’s V1-G1 subunit in glioblastoma (GBM). (**a**) Immunohistochemistry on tumor core (TC) formalin-fixed paraffin-embedded (FFPE) specimens: anti-V-ATPase-G1 (brown signal) and hematoxylin nuclear staining (blue signal). Images were captured at 20× magnification. Unit of measurement bar = 20 µm. (**b**) Differential ATP6V1G1 expression between GBM and normal brain tissue as viewed on the GEPIA2 platform. Box-plot shows different expression levels between GBM (163 cases) and normal brain tissues (207 cases). The red box indicates GBM, and the grey box indicates normal brain tissue. TPM: transcripts per kilobase of exon model per million mapped reads. * *p* < 0.05. (**c**) The median expression of ATP6V1G1 mRNA in GBM and normal brain tissue expressed in TPM is reported in the table.

**Figure 3 ijms-25-02743-f003:**
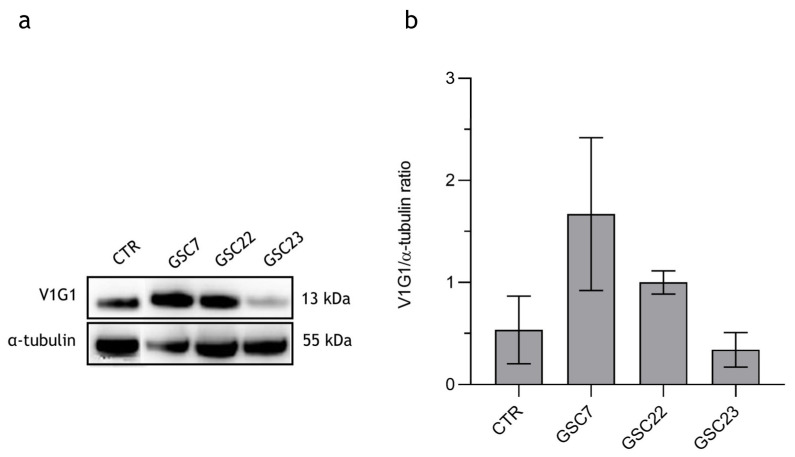
Expression of V-ATPase’s V1-G1 subunit in three glioma stem cell (GSC) lines. (**a**) Western blotting analysis of the protein V1G1. CTR: positive control colon cancer cell line HCT116. (**b**) Quantification of V1G1′s relative expression as a V1G1/α-tubulin ratio. Data are reported as the mean of three independent experiments ± S.D.; the statistical analysis was performed by comparing the samples’ ratios to the one of the control (HCT116).

**Figure 4 ijms-25-02743-f004:**
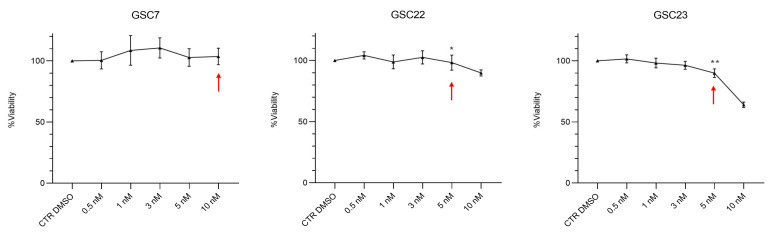
Bafilomycin A1 (Baf) dose–response curves performed on glioma stem cell (GSC) lines by a Cell Counting Kit-8 (CCK-8) assay. GSC lines were treated for 48 h with nanomolar doses of Baf (0.5, 1, 3, 5, and 10 nM). Data are represented as the mean percentage of three independent experiments ± S.D. compared to the controls (CTR DMSO, vehicle control), arbitrarily set to 100%. The statistical analysis was performed on raw data. *: *p* value < 0.05, **: *p* value < 0.01. Red arrows indicate the sub-lethal dosage of Baf identified for each GSC line.

**Figure 5 ijms-25-02743-f005:**
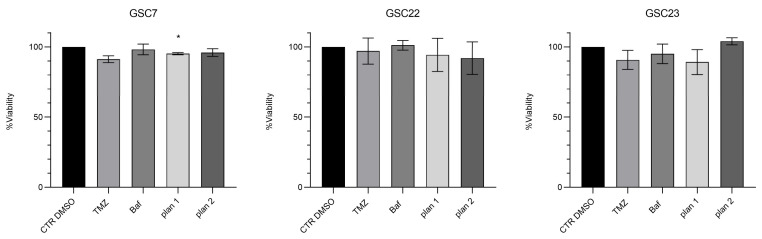
Cytotoxicity assay by Cell Counting Kit-8 (CCK-8) of glioma stem cell lines (GSC) 7, 22, and 23. GSC lines were treated for 48 h with temozolomide (TMZ, 100 μM) or a bafilomycin A1 (Baf, GSC line-specific dosage: GSC7 10 nM, GSC22 5 nM, and GSC23 5 nM) and in combination, simultaneously (plan 1), and at deferred times (24 h Baf and then TMZ addition, plan 2). Data are represented as the mean percentage of three independent experiments ± S.D. compared to the controls (CTR DMSO, vehicle control), arbitrarily set to 100%. The statistical analysis was performed on raw data. *: *p* value < 0.05.

**Figure 6 ijms-25-02743-f006:**
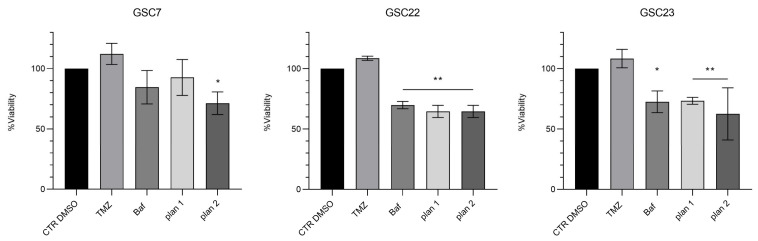
Cytotoxicity assay by Cell Counting Kit-8 (CCK-8) of glioma stem cell lines (GSC) 7, 22, and 23. GSC lines were treated for 48 h with temozolomide (TMZ, 100 μM) or bafilomycin A1 (Baf, 20 nM) and in combination, simultaneously (plan 1), and at deferred times (24 h Baf and then TMZ addition, plan 2). Data are represented as the mean percentage of three independent experiments ± S.D. compared to the controls (CTR DMSO, vehicle control), arbitrarily set to 100%. The statistical analysis was performed on raw data. *: *p* value < 0.05, **: *p* value < 0.01.

**Figure 7 ijms-25-02743-f007:**
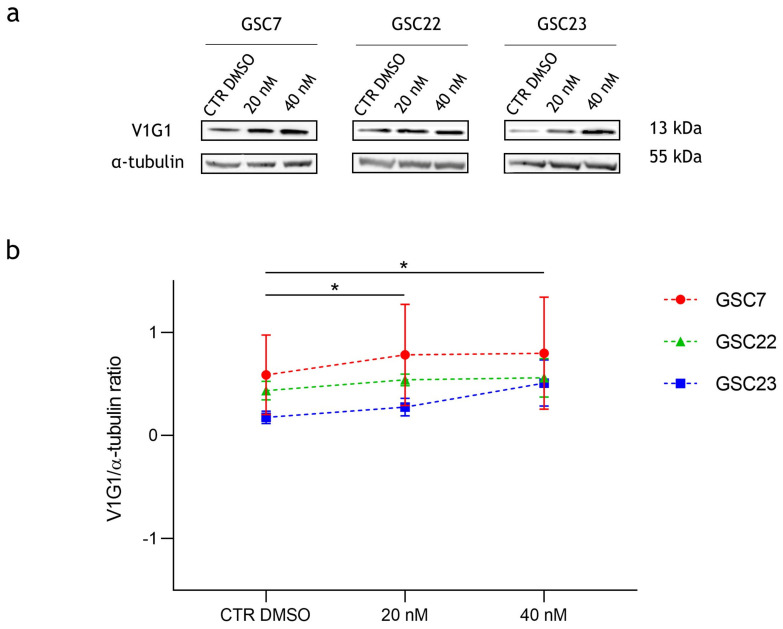
Expression of V-ATPase’s V1-G1 subunit in glioma stem cell (GSC) lines after 48 h of treatment with bafilomycin A1 (20 nM and 40 nM). (**a**) Western blotting analysis of the protein V1G1; α-tubulin was used as a loading control. CTR DMSO represented the untreated condition (cell treated with the vehicle DMSO). (**b**) Quantification of V1G1′s relative expression as V1G1/α-tubulin ratios. Data are reported as the mean ± S.D. of three independent experiments. The statistical analysis was performed by comparing both treatments’ ratios to the one of CTR DMSO and by comparing the ratios between the treatments. *: *p* value < 0.05.

**Figure 8 ijms-25-02743-f008:**
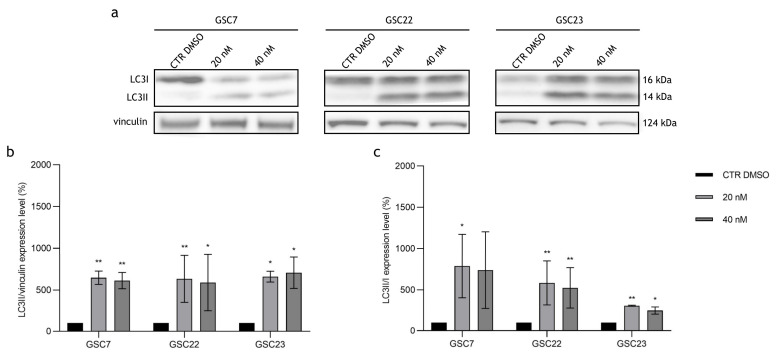
Expression of autophagy-associated proteins in glioma stem cell (GSC) lines after 48 h of treatment with bafilomycin A1 (20 nM and 40 nM). (**a**) Western blotting analysis of the proteins LC3I and LC3II; vinculin was used as a loading control. CTR DMSO represented the untreated condition (cell treated with the vehicle DMSO). (**b**) Quantification of LC3II’s relative expression as a percentage of LC3II/vinculin ratios. (**c**) Quantification of LC3II’s relative expression as a percentage of LC3II/I ratios. Data are reported as the mean of three independent experiments ± S.D. The statistical analysis was performed by comparing the samples’ ratios to the ones of CTR DMSO, arbitrarily set to 100%. *: *p* value < 0.05, **: *p* value < 0.01.

**Figure 9 ijms-25-02743-f009:**
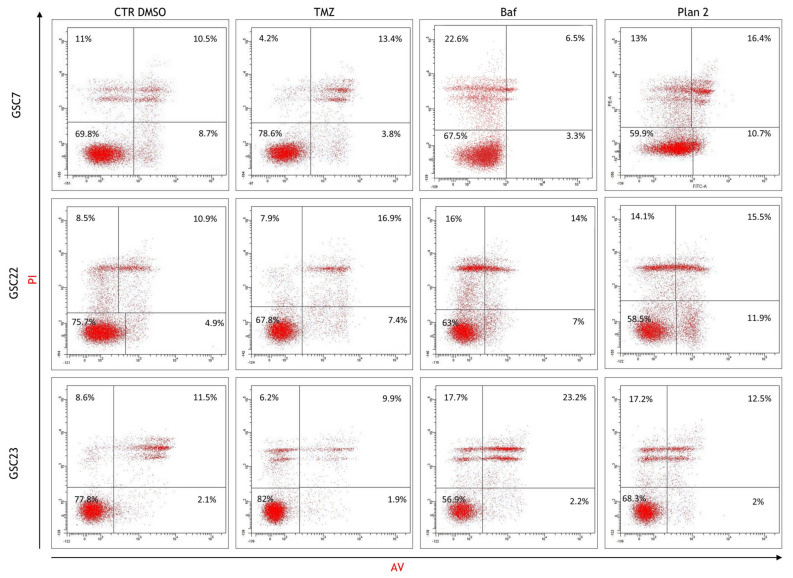
Assessment of cellular death after the treatments, using flow cytometry. Dot plots of the glioma stem cell (GSC) lines in the untreated condition (cell treated with the vehicle DMSO) and after 48 h of treatment with temozolomide (TMZ, 100 μM) and bafilomycin A1 (Baf, 20 nM), given alone and in combination at deferred times (plan 2, as described before). The horizontal axis shows Annexin V- Fitc staining (AV) and the vertical axis Propidium Iodide staining (PI). Viable cells are in the lower-left quadrant (double negative); early-apoptotic cells are in the lower-right quadrant (PI-negative and Annexin V-positive); late-apoptotic/secondary necrotic cells are in the upper-right quadrant (double positive); necrotic cells are in the upper-left quadrant (PI-positive and Annexin V-negative). The mean percentages of the cells counted in each condition are reported as well. In each quadrant, the mean percentage of the events of three independent experiments is reported.

**Figure 10 ijms-25-02743-f010:**
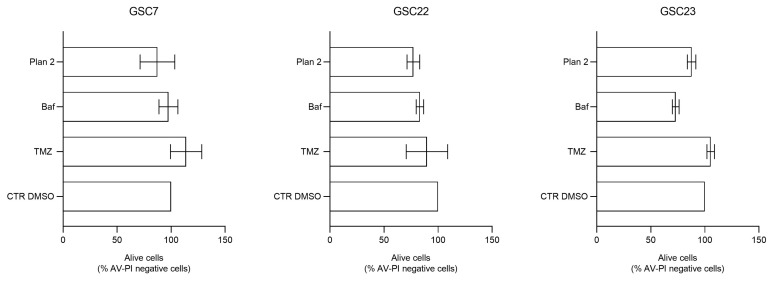
Percentage of the alive cells’ portion (cells negative for Annexin V—AV and Propidium Iodide—PI staining) after the treatments, using flow cytometry. Alive cells of each condition are compared to the control (CTR DMSO, vehicle control), arbitrarily set to 100%. Data are represented as the mean percentage of three experiments ± S.D. The statistical analysis was performed on raw data.

**Table 1 ijms-25-02743-t001:** *MGMT* promoter methylation levels in tumor core (TC) bulks and matched glioma stem cell lines (GSC) derived from three patients.

Patient	% of Methylation		Status
	TC	GSC	
**7**	1%	10%	−
22	17%	78%	+
23	38%	41%	+

The cut-off for hyper-methylation was set to ≥10%. Symbol + indicates hyper-methylation; symbol − indicates hypo-methylation.

## Data Availability

The data used to support the findings of this study are available from the corresponding author upon request.

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
