# Peer review of "Vacuolar Proton-Translocating ATPase May Take Part in the Drug Resistance Phenotype of Glioma Stem Cells"

_ijms, 2024, doi:10.3390/ijms25052743_

Round 1
Reviewer 1 Report
Comments and Suggestions for Authors
In the manuscript “V-ATPase proton pump may take part to drug-resistance phenotype of glioma stem cells”, the authors discussing about the role of V-ATPase in glioblastoma stem cells drug resistance. The paper is interesting in the way of point outing other possible drug resistance mechanism conserved in glioblastoma stemness. However, there are few concerns for this reviewer that needs to be addressed before the manuscript can be published.
1. The authors claimed that they have checked the localization of proton pump by immunofluorescence and these data should be shown.
2. There is no data showing the IC50 Baf in those cell lines. I would recommended show the IC50 of Baf in the GSC7, GSC22 and GSC23 cell lines.
3. Fig 8 western blot is not convincing, and the loading control vinculin is not even the wells.
4. I would recommend V ATPase knock down studies to confirm its role in drug resistance.
Reviewer 2 Report
Comments and Suggestions for Authors
Although the manuscript’s title is: “V-ATPase proton pump may take part to drug-resistance phenotype of glioma stem cells” the authors conclude: that their study questions previous studies where the combination of bafilomycin A1 (Baf), a V-ATPase inhibitor, as an enhancer of temozolomide (TMZ) was effective, because Baf did not overcome TMZ resistance in patient derived glioma stem cells (GSC). Moreover, nanomolar dosages of Baf blocked the autophagy process and cellular necrosis, which further questions the benefit of this combination.
Although the outcome of the study is negative it may be of help to design further studies to overcome drug resistance, especially because the underlying mechanisms and by that possible targets for combination therapy are well described.
My only question is, whether the concentrations used in the in vitro model are relevant for the in vivo situation during therapy. This should be further explained.
